# An Assessment of Government Capacity Building to Restrict the Marketing of Unhealthy Food and Non-Alcoholic Beverage Products to Children in the Region of the Americas

**DOI:** 10.3390/ijerph18168324

**Published:** 2021-08-06

**Authors:** Sofía Rincón-Gallardo Patiño, Fabio Da Silva Gomes, Steven Constantinou, Robin Lemaire, Valisa E. Hedrick, Elena L. Serrano, Vivica I. Kraak

**Affiliations:** 1Department of Non-Communicable Diseases and Mental Health, Pan American Health Organization, World Health Organization, Washington, DC 20037, USA; gomesfabio@paho.org (F.D.S.G.); constanste@paho.org (S.C.); 2Center for Public Administration and Policy, School of Public and International Affairs, Virginia Polytechnic Institute and State University, Blacksburg, VA 24061, USA; rlemaire@vt.edu; 3Department of Human Nutrition, Foods, and Exercise, College of Agriculture and Life Sciences, Virginia Polytechnic Institute and State University, Blacksburg, VA 24061, USA; vhedrick@vt.edu (V.E.H.); serrano@vt.edu (E.L.S.); vivica51@vt.edu (V.I.K.)

**Keywords:** food marketing, children, nutrition policy, government regulation, capacity-building, obesity, region of the Americas

## Abstract

The Pan American Health Organization (PAHO) Strategic Plan 2020–2025 committed to reduce children’s consumption of energy-dense nutrient-poor food and beverage products high in fat, sugar and salt (HFSS) and promote healthy eating patterns to reduce malnutrition in all forms. This paper describes the capacity-building needs in PAHO’s Member States to restrict the marketing of HFSS food and beverages to children. We asked Ministries of Health officials or national institutes/departmental representatives (*n* = 35) to complete a 28-item web-based survey (January to July 2020). Capacity-building needs were assessed using an adapted version of the World Health Organization’s government capacity-building framework with three modules: public health infrastructure, policies and information systems. Notable achievements for the PAHO’s Plan of Action were identified. State representatives reported strong infrastructure and information systems; however, policy improvements are needed to increase comprehensive national responses. These include using a constitutional health and human rights approach within the policies, policies that document conflict of interest from non-state actors, and strengthening regulatory oversight for digital media platforms. These findings provide baseline data and we suggest priorities for further action to strengthen national governments’ capacity-building and to accelerate the development, implementation, and monitoring systems to restrict the marketing of HFSS food and non-alcoholic beverages to children in the region of the Americas.

## 1. Introduction

Noncommunicable diseases (NCDs) and their risks factors are the leading causes of morbidity and disability globally, representing 70% of deaths worldwide [1]. NCDs account for 80.7% of all deaths in the 35 countries in the region of the Americas, including countries in North, Central, and South America and the Caribbean [1]. Overweight and obesity are risk factors for children, adolescents and adults and are associated with diet-related NCDs including type 2 diabetes, cardiovascular disease and cancer, which are all major public health problems [2]. The combined overweight and obesity prevalence rates in the region of the Americas is 62% for adults, 25% for adolescents and 20% for children [2].

Between 2010 and 2020, United Nations (UN) agencies issued several technical advisory documents that recommended Member States from the World Health Organization (WHO) protect children from the marketing of high fat, sugar and salt (HFSS) food and beverage products to reduce obesity and NCD risks. This paper will use the UN’s definition of *children*, which includes individuals from birth up to 18 years [3]. In May 2010, the 193 Member States (national governments) unanimously endorsed the World Health Assembly (WHA) Resolution 63.14 to restrict the marketing of HFSS food and non-alcoholic beverages to children as part of a broader call to action by Member States to address the growing obesity and NCD pandemics [4].

In 2011, Resolution WHA 63.14 was adopted by the UN General Assembly, advising Member States worldwide to implement action-oriented policies to prevent NCDs [5]. The WHO developed an NCD Progress Monitoring framework in 2012 with 10 indicators, which was updated and expanded to reflect cost-effective “best buys” [6]. This framework included a metric for Member States to reduce unhealthy diets (indicator 7), which integrated the WHO’s 2010 recommendations to restrict the marketing of HFSS food and non-alcoholic beverages to children [7].

In 2015, the UN released 17 Sustainable Development Goals (SDGs) that encourage Member States, the private sector, and civil society to address unhealthy diets as part of a broad agenda to achieve by 2030 [8]. SDG 3 is to “Ensure healthy lives and promote wellbeing for all at all ages” [9]. The WHO encouraged Member States to adopt a whole-government approach, including collaboration with non-state actors to implement and enforce adequate policies, strategies and actions to achieve the 169 targets for the 17 goals. Non-state actors include civil society organizations, academic institutions and private-sector actors such as industry and businesses. 

In the region of the Americas, several guidance documents have been issued to enable Member States to develop and strengthen policies, actions and plans to protect children’s exposure to marketing practices that promote HFSS food and non-alcoholic beverages. The PAHO released recommendations for the region on the marketing of food and non-alcoholic beverages to children [10], a Plan of Action for the Prevention and Control of NCDs [11], a Plan of Action for the Prevention of Obesity in Children and Adolescents [12], and the 2020–2025 Strategic Plan committed to reducing the consumption of energy-dense nutrient-poor HFSS products marketed to children to help reduce malnutrition in all forms [13]. 

Successful strategies to prevent and control obesity and NCDs rely on governments’ effective capacity to implement and enforce international agreements and treaties, national plans and comprehensive policies, including laws and regulations [14]. Evidence shows that a lack of adequate governance capacity is a major barrier for governments to develop and enforce effective policies to control obesity and NCDs [15] and achieve SDG 3 and global nutrition goals by 2030 [16]. In the region of the Americas, Member States have developed strong NCD policies [17]; however, there is limited information about countries’ capacity-building needs to monitor, enforce, develop and enhance comprehensive policies to achieve local, regional and global nutrition targets. 

This study’s purpose is to assess the capacity-building needs in the region of the Americas to fully implement the 2010 Resolution WHA 63.14 to restrict the marketing of HFSS food and non-alcoholic beverages to children. First, we describe two conceptual frameworks used to map Member States’ capacity-building needs as well as the food and beverage marketing practices used by industry actors. Thereafter, we describe a cross-sectional online survey conducted in 2020 to document the policy and regulatory environment in the region of the Americas. Finally, we identify data from policies, strategies and actions implemented throughout the region that will serve as a baseline to strengthen the development, implementation and monitoring of progress to regulate the marketing of food and non-alcoholic beverages to children in the future.

## 2. Materials and Methods

### 2.1. Capacity-Building Framework

Diverse capacity-building definitions, model and frameworks have been used in public health to enhance the workforce, strengthen institutional mechanisms, facilitate intersectoral engagement between stakeholders, expand intellectual and economic resources and assess institutional performance [18,19,20,21,22]. The WHO defines capacity-building as “the development of knowledge, skills, commitment, structures, systems and leadership to enable effective action” [23]. Many capacity-building frameworks are intended to enhance the workforce to support qualified professionals. This study used an adapted version of the WHO’s government capacity-building framework to assess national capacity for the prevention and control of NCDs [24] (Figure 1). The framework depicts three elements, including: (1) public health infrastructure (i.e., organizational development, workforce, multisectoral collaboration and human and financial resources); (2) policy efforts (i.e., status of policies and action plans); and (3) information systems (i.e., management quality assessment, monitoring, surveillance and surveys). 

An Integrated Marketing Communications (IMC) framework [25] was also included in the study to identify and assess the specific marketing practices addressed by Member State’s action plans, policies, regulations, laws and strategies to restrict the marketing of food and non-alcoholic beverages to children in the region of the Americas. The IMC framework describes how different marketing strategies and techniques are used to target children and adolescents through diverse settings, media channels and platforms, which influence their diet-related cognitive, behavioral and health outcomes [25] (Figure 2). 

### 2.2. Data Collection

Through the PAHO regional office, we sent an email to each of the 35 country offices in the region of the Americas requesting that a Ministry of Health official or national institute/departmental representative respond a 28-item web-based survey, between 14 January 2020 and 20 August 2020. Respondents needed to be working with government units, branches or departments responsible for addressing the diet- and health-related NCDs in their countries or have comprehensive knowledge on the topic to complete the survey. Each respondent received unique details to access an online platform where they could answer the survey. To validate and verify the answers, respondents were asked to submit supporting documentation (i.e., official national documents or reports). 

### 2.3. Online Survey

A 28-item web-based survey (Appendix A) was developed and adapted from the WHO Global Survey to Asses National Capacity for the Prevention and Control of NCDs [24] by technical experts on policies to restrict the marketing of HFSS foods and non-alcoholic beverages to children. The PAHO’s Plan of Action for the Prevention of Obesity in Children and Adolescents [12] was also considered in the development of the survey. 

The web-based survey was available for completion in English and Spanish. It consisted of a general information section and three modules to assess capacity-building. Module 1 addressed public health infrastructure, included questions relating the staff, funding and synergies from the unit or division responsible for addressing the restriction of marketing of food and non-alcoholic beverages to children. Module 2 addressed policy efforts, the presence of specific components addressed in policies, strategies and action plans, including national health targets, entities or bodies that oversee and enforce, accountability mechanisms and cross-border marketing strategies. Module 3 addressed information systems, and monitoring and surveillance activities.

### 2.4. Data Analysis

The units of analysis of the study are countries. Data were downloaded and coded directly from the web-based platform to an Excel-readable file and were cleaned to ensure consistency with responses within questions across each respondent country. Responses of “do not know” or unanswered were treated equally.

## 3. Results

Responses were received by 17 of the 35 Member States in the Region of the Americas, representing a response rate of 48.6%. However, the web-based survey was fully completed by 16 countries. The U.S. government representative responded with a letter stating that it had made significant efforts and progress in the area but did not complete the online survey. Table 1 and Table 2 list the Member State respondents who participated in the survey. 

### 3.1. Public Health Infrastructure

Public health infrastructure addressed organizational development, workforce, multisectoral collaboration and human and financial resources (Table 1). The web-based survey revealed that countries (*n* = 11) had available a unit, branch or department responsible to address the marketing of HFSS food and non-alcoholic beverages to children. All respondents reported having at least one full-time technical or professional staff member working within the unit, branch or department; fewer countries (*n* = 7) reported having more than five full-time technical professional staff members.

Funding earmarked in the national government’s budget to support activities to address restrictions in marketing HFSS food and non-alcoholic beverages to children was reported by eleven (*n* = 11) countries, and six (*n* = 6) reported having budgetary funding for all measured components (i.e., policy design and implementation, surveillance, monitoring and evaluation, capacity-building and research). The component with the most allocated funding was surveillance, monitoring and evaluation, reported by ten (*n* = 10) counties, followed by policy design and implementation, reported by nine (*n* = 9), capacity-building by seven (*n* = 7), and research by six (*n* = 6) countries.

Partnerships or multisectoral collaborations with other institutions, companies, business alliances, industry trade associations or individuals to enhance policies that address restrictions for marketing HFSS food and non-alcoholic beverage products to children was reported by twelve (*n* = 12) countries. Partnerships with other government ministries were reported by eleven (*n* = 11), with UN agencies by ten (*n* = 10), with civil society organizations by nine (*n* = 9) and with private-sector firms by two (*n* = 2) countries.

The evidence sources used to develop, implement or modify policies were reported to have been taken from government guidelines by all countries, from UN documents by nine (*n* = 9), and from scientific evidence by six (*n* = 6) countries. 

Eleven (*n* = 11) countries reported having leaders, policymakers, champions or advocacy organizations that influenced the development of strategic directions, motivated staff, and aligning goals to implement policies effectively; eight (*n* = 8) reported having a government representative, eight (*n* = 8) reported having a civil society representative, four (*n* = 4) having a UN representative, and three (*n* = 3) countries reported having an academic representative.
ijerph-18-08324-t001_Table 1Table 1Capacity-building public health infrastructure characteristics to restrict the marketing of HFSS food and non-alcoholic beverages in countries in the region of the Americas.Country^1^ Funding ^2^ Partnerships *^3^ Evidence Resources^4^ LeadersPolicy Design and ImplementationSurveillance SystemsCapacity-BuildingResearchOther Government Ministries UN AgenciesAcademic InstitutionsCivil Society OrganizationsPrivate-Sector Firms or OrganizationsGovernmentUN DocumentsScientific EvidenceAcademiaGovernmentCivil Society OrganizationsUN AgencyBahamasNoNoNoNoNoNoNoNoNo-------BrazilYesYesYesYesYesYesYesYesNoYesYesYesNoYesYesNoCanadaNoYesNoNoYesYesYesYesNoYesYesYesNoNoYesNoChileYesYesYesNoYesNoYesYesNoYesYesYesNoYesYesNoCosta RicaNoYesNoNoNoNoNoNoNoYesYesNoNoYesNoNoCubaYesYesYesYesYesYesYesYesNoYesYesNoNoYesNoNoDominican RepublicYesYesYesYesYesYesYesYesYesYesNoNoNoNoYesNoEcuadorYesYesNoNoYesYesYesYesNoYesYesYesNoNoYesNoGuatemalaNoNoNoNoYesYesYesYesNoYesYesNoYesYesYesYesGuyanaYesYesYesYesYesYesNoNoNoYesNoNo----HaitiNoNoNoNoNoNoNoNoNo-------HondurasNoNoNoNoNoNoNoNoNoYesNoNo----MexicoYesYesYesYesYesYesYesYesNoYesYesYesYesYesYesYesParaguayYesNoNoNoYesYesYesYesYesYesNoNoYesYesYesYesUruguayNoNoNoNoYesYesYesNoNoYesYesYes----VenezuelaYesYesYesYesYesYesYesNoNoYesNoNoNoYesNoYesResponses were categorized as *Yes* to indicate a positive and *No* to indicate negative actions within the specific items related to capacity-building public health infrastructure. The questions asked for each item (marked with superscripts) were as follows: ^1^
**Funding sources:** Indicate with yes or no wether there is funding allocated in the national government’s budget to support any policies to address the marketing of HFSS food and non-alcoholic beverage products to children in your country; ^2^
**Partnerships**: Select the options that apply (Other government ministries, United Nations agencies, Academia, Private-sector firms) on partnerships or multisectoral collaborations to address the marketing of HFSS food and non-alcoholic beverage products to children in your country; ^3^
**Evidence resources**: Indicate below the type of evidence resources used to develop, implement, or modify policies, strategies, or actions to address the marketing of HFSS food and non-alcoholic beverages to children in your country; and ^4^ **Leaders:** Indicate if there are effective leaders (policymakers, advocacy champions, public organizations) who create strategic directions, motivate staff, and align goals to implement policies, strategies, or actions to address the marketing of HFSS food and non-alcoholic beverages to children in your country. Please specify. * Other government ministries (i.e., Ministry of Education, Ministry of Finance, and Ministry of Social Welfare); HFSS: high in fat, sugar, and salt; UN: United Nations agencies (i.e., The World Bank, Pan-American Health Organization/World Health Organization (WHO), Food and Agricultural Organizations (FAO), and United Nations Children’s Fund (UNICEF)); Academia (i.e., research centers or universities); Private-sector firms or organizations (i.e., food and beverage manufacturers, restaurants, retailers, food service companies, entertainment and media companies, corporate foundations, business alliances, and industry trade organizations).

### 3.2. Policy Efforts

Policy efforts addressed the status of policies, strategies and actions to restrict the marketing of HFSS food and non-alcoholic beverages to children, including the presence or absence of national targets and enforcement mechanisms (Table 2). While most of the countries and territories included in their national constitution explicit language about children’s right to food and health (*n* = 13), only eleven (*n* = 11) countries reported prioritizing a rights-based approach for policies to address the marketing of HFSS food and non-alcoholic beverages to children. Two countries reported prioritizing the issue on their national agenda, and they did not have a specific government action but reported plans to take further steps. Nine (*n* = 9) countries reported having a government policy, strategy or action in place. Few countries responded that they had a measurable and time-scaled objective for a national target to restrict or reduce the marketing of HFSS food and non-alcoholic beverages to children under the implementation of the SDG Goal 3 to Ensure healthy lives and promote well-being for all the ages.

Table 3 shows the mandatory (i.e., required) and voluntary (i.e., free will) components within the government policies in place reported by the countries. Among those countries that had a policy, strategy or action (*n* = 9), product design and packaging was the marketing strategy most prevalent as a mandatory policy (*n* = 6) whereas sponsorship was the least prevalent policy (*n* = 2). Schools were the most common protected setting (*n* = 7), while digital and media platforms were addressed the least (*n* = 2) by countries. 

From the countries implementing a policy, strategy or action (*n* = 9), all had used national dietary guidelines as the threshold to limit or restrict the marketing of HFSS food and non-alcoholic beverages to children; six (*n* = 6) countries had used the PAHO’s nutrient profile model; six (*n* = 6) countries used independent criteria; and five (*n* = 5) countries used food and beverage categories. Results also showed that different nutrition criteria were used in distinct policies, strategies and actions within the same country.

### 3.3. Information Systems

Information systems addressed accountability mechanisms such as management quality assessment, monitoring, surveillance systems and surveys (Table 4). The survey found that accountability mechanisms were used to enforce policies by nine (*n* = 9) countries and that seven (*n* = 7) countries used financial penalties as an enforcement measure. Governmental entities responsible for overseeing and enforcing policies were reported by twelve (*n* = 12) countries but only six (*n* = 6) addressed cross-border marketing.

Governments face several barriers to increase government capacity-building to restrict the marketing of HFSS foods and non-alcoholic beverages to children in the region of the Americas. To overcome these difficulties, we suggest priorities for further action based on the results (Table 5).
ijerph-18-08324-t002_Table 2Table 2Capacity-building policy efforts to restrict the marketing of HFSS food and non-alcoholic beverages in countries in the region of the Americas.CountryChildren’s Rights to Food and Health in the National ConstitutionPolicies, Strategies or ActionsBody, Entity, or Institution in Charge of EnforcementAccountability MeasuresCross-Border MeasuresMarketing StrategiesMedia Channels, Platforms and SettingsNutrition CriteriaBahamas--------BrazilYesYesNational Health Surveillance Agency (Anvisa)Ministry of Justice,Public Ministry and National Consumer Protection SystemFinesPublic complaintsVerbal warningsMedia reportsNoCartoon characters and celebritiesDirect marketingPremium offersProduct design and packagingBroadcastDigital and socialmediaFood retailers and restaurantsMobile and digital devicesOutdoors and transportationSchoolsWebsitesNational dietary guidelinesIndependent criteriaPAHO nutrient profile modelFood and beverage categoriesCanada--------ChileYesYesMinistry of HealthFinesYesCartoon characters and celebritiesDirect marketingPoint of salePremium offersProduct design and packagingSponsorshipBroadcastCommunity, sports and special eventsDigital and socialmediaFood retailers and restaurantsMobile and digital devicesOutdoors and transportationSchoolsWebsitesNational dietary guidelinesIndependent criteriaCosta RicaYes-------CubaYesYesNational Standardization OfficeMinistry of Public HealthFinesMedia reportsYesPoint of saleProduct design and packagingSponsorshipBroadcastSchoolsNational dietary guidelinesFood and beverage product categoriesDominican RepublicYesYesMinistry of Public Health and Social AssistanceAttorney General’s Office of the Republic, Ministry of the Interior and PoliceFinesNoNot specifiedOutdoors and transportationSchoolsNational dietary guidelinesIndependent criteriaPAHO NPMFood and beverage product categoriesEcuadorYesYesNational Agency for Health Regulation, Control and Surveillance (ARCSA)FinesYesDirect marketingPoint of saleProduct design and packagingProduct placementFood retailers and restaurantsSchoolsNational dietary guidelinesIndependent criteriaPAHO NPMFood and beverage product categoriesGuatemalaYes-------GuyanaYes-------Haiti--------HondurasYes-------MexicoYesYesFederal Consumer Protection AgencyFederal Commission for Protection against Health Risks (COFEPRIS)
YesBrandingCartoon characters and celebritiesPremium offersProduct design and packagingProduct placementSponsorshipBroadcastFood retailers and restaurantsSchoolsNational dietary guidelinesIndependent criteriaPAHO NPMFood and beverage product categoriesParaguayYesYesNational Institute of Food and Nutrition (INAN)Ministry of Public Health
NoN/AN/ANational dietary guidelinesUruguayYesYesNone
NoBrandingDirect marketingPoint of saleSchoolsNational dietary guidelinesPAHO NPMVenezuelaYesYesMinistry of the Popular Power for HealthFinesYesProduct design and packagingProduct placementFood retailers and restaurantsNational dietary guidelinesIndependent criteriaPAHO NPMResponses were categorized as *Yes* to indicate a positive and *No* to indicate negative actions within the specific items related to capacity-building policy efforts. N/A = not available; “-” = no data reported; HFSS: high in fat, sugar, and salt; PAHO NPM: Pan American Health Organization’s nutrient profile model.ijerph-18-08324-t003_Table 3Table 3Marketing strategies, channels, platforms and settings covered in countries in the region of the Americas that have enacted policies, strategies or actions to restrict the marketing of HFSS food and non-alcoholic beverages.
**Brazil****Chile****Cuba****Dominican Republic****Ecuador****Mexico****Paraguay****Uruguay****Venezuela****Marketing Strategies**Brandingo--o-•N/A••Cartoon characters and celebrities••-o-•N/A--Direct Marketing••-o•oN/A••Point of Saleo••o•oN/A••Premium offers••-o-•N/A--Product design and packaging •••o••N/A-•Product Placemento--o••N/A-•Sponsorshipo•-o-•N/A--**Media channels, platforms, and settings**Broadcast•••o-•N/A-oCommunity, sports, and special events-•o--oN/A-oDigital and social media••o--oN/A-oFood retailers and restaurants••-o-•N/A-•Mobile and digital devices••o--oN/A-oOutdoors and transportation ••-•-oN/A-oSchools ••••••N/A•oWebsites••-o-oN/A-o* N/A= not available; “-” = no data reported; • = Mandatory; o = Voluntarily.
ijerph-18-08324-t004_Table 4Table 4Capacity-building information systems to restrict the marketing of HFSS food and non–alcoholic beverages in countries in the region of the Americas.CountryMonitoring SystemEntities Responsible for Collecting Data and Reporting ResultsSurveillanceSurveysIndustry and Media ReportsGovernmentIndustry Corporations and Trade AssociationsResearch InstitutionsPublic Interest and Civil Society OrganizationsInvestigative Journalists in the MediaBahamas--------BrazilNoYesNoYesNoNoYesNoCanadaNoNoYesYesNoNoNoNoChileYesNoNoYesNoYesNoYesCosta Rica--------CubaNoNoYesNoNoYesNoNoDominican RepublicNoNoYesYesYesNoNoYesEcuadorYesNoNoYesNoYesNoNoGuatemala--------GuyanaNoYesNoYesNoNoNoNoHaiti--------Honduras--------MexicoNoYesNoYesNoYesNoNoParaguayNoYesNoYesNoNoNoNoUruguayNoYesNoYesNoNoNoNoVenezuelaNoYesNoYesNoYesNoNoResponses were categorized as *Yes* to indicate a positive and *No* to indicate negative actions within the specific items related to capacity-building information systems; “-“ = no data reported.
ijerph-18-08324-t005_Table 5Table 5Priorities for further action to increase government capacity-building to restrict the marketing of HFSS foods and non-alcoholic beverages to children in the region of the Americas.**Public Health Infrastructure**FundingLeverage resources of funding for policy design and implementation, surveillance and monitoring, capacity-building and research.Multisectoral collaborationsAdoption of mechanisms to report conflict of interest to hold stakeholders accountable and ensure good governance and the effectiveness of collaborations.ResourcesScientific evidence and UN documents without conflict of interest should be used further as main sources to design, develop and implement policies.**Policy Efforts**Policy effortsUse constitutional rights to open an avenue and claim remedies for actions to protect health-related rights by nations. Align national and international agendas by integrating the UN goals and plans to facilitate policy coherence between policies and across sectors.Policy componentsInclude a broader scope to include the diverse marketing strategies, techniques, media channels, platforms and diverse settings in order to develop comprehensive policies. Priority should be given to implementing more comprehensive policies, including digital marketing and media, which exert influence on children.Use PAHO’s nutrient criteria across the region of the Americas to accelerate coordinated progress and reduce trade barriers across national borders.Integrate cross-border marketing into multilateral, regional and unilateral trade policies to avoid weakening efforts; strengthen efforts to include public health interests, ensuring policy coherence and increasing positive effects. Policy implementationReinforce and improve financial coercive measures, other complaint mechanisms and fiscal powers to hold industry to account for its food and beverage marketing practices.**Information systems**Information systemsCivil society and academic researchers should contribute to monitoring and evaluation using standardized and systematic analytical tools to address gaps and inform policymakers.Acronyms: PAHO: Pan American Health Organization; UN: United Nations.

## 4. Discussion

This study assessed government capacity-building in the region of the Americas to restrict the marketing of HFSS foods and non–alcoholic beverages to children. The research provides baseline data to strengthen the capacity-building of PAHO Member States to achieve local, regional and global nutrition targets. The results identify major policy gaps, discuss regional plans of action and suggest priorities for further action. Despite findings that demonstrate countries have adequate human resources to facilitate policies to address the persuasive marketing of HFSS foods and non-alcoholic beverages to children, insufficient resources were reported that limited States’ capacity in research, design, implementation, surveillance and monitoring efforts across the region. A lack of funding is a major barrier to achieve adequate public health infrastructure to restrict the marketing of unhealthy food and non-alcoholic beverage products to children. These results concur with previous research that shows that infrastructure investments (e.g., technical assistance and workforce as well as research, monitoring and evaluation plans) effectively improves the government’s capacity to enforce and enact national policies [14,15,16,17,18,19,20,21,22,23,24,25,26]. In addition, the absence of budgetary resources is a common limitation that negatively affects the translation of the best evidence into laws [14]. Evidence-based policy promotes policy decisions based on the best available evidence and/or best practices that have shown to be effective [27]. The assessed countries reported that government documents were the most used source in the development and implementation of policies. Nonetheless, scientific evidence and UN documents without conflicts of interest should be used as main evidence sources due to their commitment to providing technical guidance based on the best available evidence and practices. 

The multisectoral collaborations with non-state actors (e.g., academia, civil society and UN organizations) demonstrated progress toward the PAHO’s strategic plans [12,13], which encouraged Member States to use an integrated multisectoral approach to implement evidence-based policy and improve capacity-building efforts. Thus, understanding the motives and actions of the involved stakeholders is necessary to hold stakeholders accountable, avoid conflict of interest, and ensure good governance and effectiveness of such collaborations [25]. Uncoordinated actions and industry opposition were common barriers to creating unbiased evidence-based policies. 

Most of the countries in the region of the Americas have ratified international UN human rights agreements, and have included provision in their national Constitutions explicit language to protect children’s rights to adequate nutritious food and health. However, not all reported prioritizing a child rights approach in their policies to restrict the marketing of HFSS food and non-alcoholic beverages to children. Constitutional rights are a viable strategy to claim actions to guarantee and protect human rights [28]. This approach improves Member States’ capacity-building and could be used by civil society organizations to hold state actors accountable as seen in Colombia [29] and Mexico [30]. The Colombian non-governmental organization “Educar Consumidores” defeated a regulatory order aimed to ban a broadcast campaign that highlighted the health risks and sugar content of sugar-sweetened beverages, using arguments based on the health rights of the consumers [29]. In 2015, a Mexican consumer rights organization “El poder del Consumidor” urged the government to eliminate an ineffective front-of-pack (FOP) nutrition labeling policy, stating that it violated consumers’ rights [30]. In 2020, Mexico enacted a law that required warning FOP labels using expert and PAHO/WHO recommendations [31]. 

Although many of the countries had implemented government policies to address the marketing of HFSS food and non-alcoholic beverage products, few countries reported having incorporated national targets to achieve the UN or WHO global targets. The absence of goals, targets and policy coherence is a common challenge that undermines actions taken at multiple government levels rather than supporting more cohesive objectives [32,33]. Governments should align national and international agendas to facilitate policy coherence between policies and across sectors to leverage opportunities and accelerate response to address obesity and NCDs. 

The results of this study show that countries in the region of the Americas are taking actions to implement the PAHO’s Plan of Action for the Prevention of Obesity in Children and Adolescents; specifically, action 2 (i.e., prevent the availability of energy-dense nutrient products and sugar-sweetened beverages in schools) and action 3 (i.e., regulation of food marketing and labeling by enacting actions to protect children from its impact). Schools were the most prevalent settings, and product design and packaging was the marketing technique most frequently mentioned in policies, whereas digital marketing platforms, social media and websites were rarely addressed. Similar results have been shown in previous studies, where digital platforms and techniques were rarely found in assessed policies [34,35]. 

The absence of policies to address digital marketing and media was a major policy gap to adequately protect children from exposure to the marketing of HFSS food and non-alcoholic beverage products. The main barrier to achieve regulatory policies on digital HFSS food and non-alcoholic beverage marketing is the borderless nature of the digital world, where cross-border marketing represents a challenge for national governments. The COVID-19 pandemic has accentuating the use of digital platforms (e.g., online food purchases) that affects eating behaviors with direct-to-consumer marketing that promotes the higher consumption of HFSS products [36,37]. Obesity increases the risk of complications from COVID-19 [38], which increases the urgency to achieve the global nutrition targets (e.g., WHO, UN Decade of Action and SDGs). Member States should be more ambitious to implement comprehensive policies at a faster pace to achieve global nutrition targets, especially during the dual pandemic of obesity and COVID-19. 

The inclusion of marketing strategies that have had limited public health attention, such as sponsorship, need to be considered, since these are prominently increasing and impacting HFSS food and beverage marketing [39,40]. Efforts that are exclusively focused on a specific setting or marketing technique leave children exposed to many communication channels and platforms where HFSS products are still promoted. Changes are recommended to adopt more comprehensive actions, including digital technology and media platforms that influence children’s diet and health [35,36,37,38,39,40,41]. Member States should include a broader scope of strategies that use the IMC framework [25] to develop comprehensive policies to address the marketing of HFSS food and non-alcoholic beverages to children. 

The nutrition criteria used to identify foods and beverages appropriate for marketing to children vary widely across the region. The variety of nutrition criteria could be explained by the evolutionary progress of policies and political timing. Findings show a pattern in accordance with the most recent policies on using the PAHO’s nutrient profile model, considered as the most strict standards for government mandatory regulations [42,43,44,45,46]. Applying PAHO’s nutrient criteria across the region of the Americas has the potential to accelerate States’ capacity to coordinate progress and reduce trade barriers across national borders. However, countries should avoid reducing higher standards to lower levels for the purposes of harmonization. 

Since less than half of the 35 countries in the America’s region that responded to this survey reported an integration of cross-border marketing effects into their policies, considerable future work is needed in this area. Countries receive marketing from beyond their borders, which represents a barrier to ensure the effectiveness of national policies. International trade agreements can affect obesity and NCD risk factors by facilitating trade for energy-dense nutrient-poor products. Integration of cross-border marketing into multilateral, regional and unilateral trade policies is needed to avoid weakening efforts, and public health interest must ensure policy coherence to have positive effects [32,33,34,35,36,37,38,39,40,41,42,43,44,45,46,47,48]. 

The accountability mechanisms reported to enforce policies varied widely, with financial penalties being the most common. Previous research has encouraged governments to strengthen coercive financial measures, complaint mechanisms, and fiscal powers to hold industry to account for food and beverage marketing practices [49,50]. Experience with violations with laws in Brazil and Chile shows that coercive measures (i.e., fines) can hold the food industry to account for their practices. In 2018, McDonald’s was fined over USD $1 million in Brazil for abusive advertising directed to children by using an educational venue to encourage the consumption of McDonald’s HFSS food products [51]. In Chile, the national service for the consumer, called SERNAC, sued three multinational food manufacturers (i.e., Nestle, Kellogg’s and MasterFoods) for violating the law by using brand mascots to promote HFSS products to children, and each company paid a USD $110,000 fine [52]. 

The impact and effectiveness of the policies rely on implementation; therefore, Member States should consider coercive measures to enhance their capacity to strengthen accountability and compliance mechanisms through policies that restrict marketing of HFSS food and non-alcoholic beverages to children. Substantial efforts were found for information systems in the region of the Americas. Results showed compliance with PAHO’s Plan of Action for the Prevention of Obesity in Children and Adolescents [12]; specifically, action 5 (i.e., use national information systems to monitor and generate evidence for policy decision-making). Monitoring systems allow correction of unforeseen flaws, facilitating countries’ capacity to strengthen, develop and implement nutrition policies [48,49,50,51,52,53]. Civil society and academic researchers should contribute to the monitoring and evaluation process using previously standardized and systematic analytical tools [34,35,36,37,38,39,40,41,42,43,44,45,46,47,48,49,50,51,52,53] across countries to address gaps and inform policymakers. 

### Study Strengths and Limitations

A main strength of this research is the adapted web-based survey from the WHO that assessed the national capacity-building needs to prevent and control NCDs. This web-based survey was based on a methodology to collect information in a systematic, repeatable way across countries. Moreover, it offers a unique and useful tool that enables researchers to identify specific components, areas and elements to strengthen Member States’ capacity-building efforts for developing, implementing, monitoring, and evaluating policies to protect children from the persuasive marketing of HFSS foods and non-alcoholic beverages. Further research could use the web-based survey in other WHO regions to identify priority areas of action for this issue.

A limitation of this study was the breadth of the survey had disadvantages because it was not possible to request supporting documentation for each item. However, the combined capacity-building and IMC frameworks captured a clearer picture, which provides a broad context of the diverse policies that countries had in place. Another limitation was the low response rate (48.6%) of participating countries. However, the respondents who completed the online surveys represented a wide geographical area from the region of the Americas (i.e., North, Central and South America and the Caribbean), providing an overall panorama. While definitions were available throughout the survey to address doubts, one possible limitation and potential bias of the survey is that country representatives may have interpreted the questions differently due to various characteristics and backgrounds. Nonetheless, the country offices were requested to identify respondents with the same description to complete the survey (i.e., Ministry of Health officials or national institute/department officials working with government units, branches or departments responsible to address diet- and health-related NCDs or who had comprehensive knowledge on the topic). Responses were reviewed and validated through triangulation by cross-verification with the provided supporting documents. Future research should assess the quality and implementation of policies to determine their effectiveness. 

## 5. Conclusions

Despite notable achievements with regards to the PAHO’s Plans of Action, there remain many areas of growth and improvement for government capacity-building to restrict the marketing of HFSS food and non-alcoholic beverages in countries in the region of the Americas. Countries in the region have relatively strong public health infrastructure and information systems. However, increased policy efforts are needed to create comprehensive national responses. These include using a constitutional health and human rights approach within the policies, declaration of conflict of interest from non-state actors, and strengthening regulations on digital media platforms. These findings provide baseline data and suggest priorities for further action to strengthen capacity-building and accelerate policy development, implementation and monitoring to restrict the marketing of HFSS food and non-alcoholic beverages to children in the region of the Americas. Member States should be more ambitious to implement comprehensive policies at a faster pace to achieve local, regional and global nutrition targets in a timely manner (i.e., PAHO/WHO, UN and SDGs), especially during the dual pandemic of obesity and COVID-19. 

## Figures and Tables

**Figure 1 ijerph-18-08324-f001:**
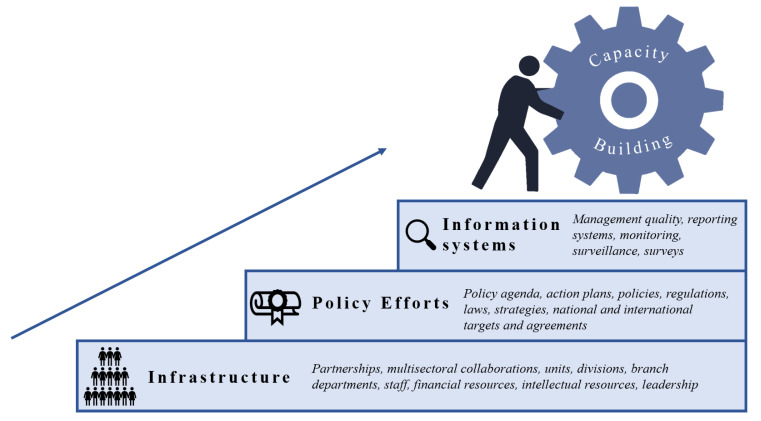
Government capacity-building framework. Adapted from World Health Organization, 2020 [24]; World Health Organization, 2006 [23]; Bergeron K, et al. BMC Public Health, 2017 [19]; Delisle H, et al. Bull WHO, 2017 [20].

**Figure 2 ijerph-18-08324-f002:**
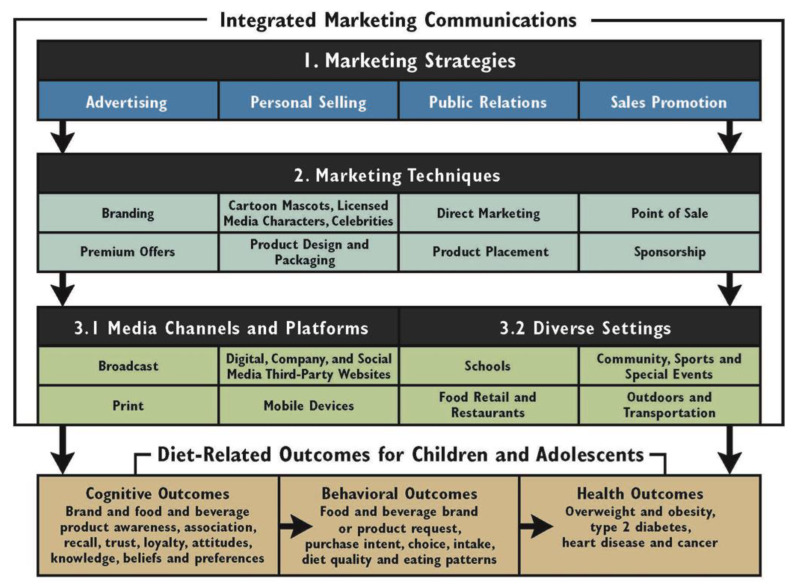
Integrated Marketing Communications Framework of marketing strategies that influence diet-related outcomes for children [25]. Reprinted with permission (figure has copyright permission from the authors).

## Data Availability

No new data were created in this study.

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
