# Peer review of "An Assessment of Government Capacity Building to Restrict the Marketing of Unhealthy Food and Non-Alcoholic Beverage Products to Children in the Region of the Americas"

_ijerph, 2021, doi:10.3390/ijerph18168324_

Round 1
Reviewer 1 Report
I am grateful for the opportunity to review the manuscript presented to me. I hope that the comments in the review would be helpful. I believe the paper is worth considering for publication, however requires minor revision.

Author Response
Thank you for taking the time to assess the manuscript titled "An assessment of the capacity-building to restrict the marketing of unhealthy food and non-alcoholic beverage products to children in the Region of the Americas." We have adressed all the concerns that you and the other reviewers raised.
Please see attachment that include a response to reviewers in which all the comments are listed in a table with our response and revisions made into the manuscript.

Reviewer 2 Report
In this manuscript, the authors conducted a web-based survey investigating the capacity-building needs to restrict the marketing of unhealthy food and non-alcoholic beverage products to children through three modules, public health infrastructure, policies, and information systems. They focused on the region of the Americas, although the biggest country, the USA, was not included in the final analysis. Nevertheless, the manuscript has some value in that it provides an assessment of the capacity-building needs of state representatives in the region of the Americas. I feel that the manuscript can be improved if the authors can also consider the barriers to achievements in the three modules, especially policy improvements. Furthermore, the first and third sentences of the first paragraph of the Introduction need references. There are also occasional typos throughout the manuscript, which should be addressed before acceptance.
Author Response

(The authors gave the same response as above.)

Reviewer 3 Report
Thanks for inviting me to review this paper investigating the capacity-building to restrict the marketing of unhealthy food and non-alcoholic beverage products to Children in the region of the Americas. Generally speaking, this paper provides a decent overview of this issue specific to the central and south American regions. However, the only concern that I might have is the scientific soundness of study design. I may have some doubts regarding the representativeness of this survey or the cohort of respondents. It may be responsible to include some descriptions of the characteristic of the respondents and acknowledge this as a potential bias in this report. Besides, there are some grammatical errors throughout the paper. Having someone proofread this paper may help to improve the correctness of the arguement.
Author Response

(The authors gave the same response as above.)

Round 2
Reviewer 3 Report
Thanks for the efforts paid to address my concerns. I have no more questions on this present version.